# Distributed Attack Modeling Approach Based on Process Mining and Graph Segmentation

**DOI:** 10.3390/e22091026

**Published:** 2020-09-14

**Authors:** Yuzhong Chen, Zhenyu Liu, Yulin Liu, Chen Dong

**Affiliations:** 1Fujian Key Laboratory of Network Computing and Intelligent Information Processing, College of Mathematics and Computer Science, Fuzhou University, Fuzhou 350116, China; yzchen@fzu.edu.cn (Y.C.); n170325003@fzu.edu.cn (Z.L.); 2Key Laboratory of Spatial Data Mining & Information Sharing, Ministry of Education, Fuzhou 350116, China; 3Key Laboratory of Information Security of Network Systems, Fuzhou University, Fuzhou 350116, China; 031703113@fzu.edu.cn

**Keywords:** process mining, attack graph, attack model, graph segmentation

## Abstract

Attack graph modeling aims to generate attack models by investigating attack behaviors recorded in intrusion alerts raised in network security devices. Attack models can help network security administrators discover an attack strategy that intruders use to compromise the network and implement a timely response to security threats. However, the state-of-the-art algorithms for attack graph modeling are unable to obtain a high-level or global-oriented view of the attack strategy. To address the aforementioned issue, considering the similarity between attack behavior and workflow, we employ a heuristic process mining algorithm to generate the initial attack graph. Although the initial attack graphs generated by the heuristic process mining algorithm are complete, they are extremely complex for manual analysis. To improve their readability, we propose a graph segmentation algorithm to split a complex attack graph into multiple subgraphs while preserving the original structure. Furthermore, to handle massive volume alert data, we propose a distributed attack graph generation algorithm based on Hadoop MapReduce and a distributed attack graph segmentation algorithm based on Spark GraphX. Additionally, we conduct comprehensive experiments to validate the performance of the proposed algorithms. The experimental results demonstrate that the proposed algorithms achieve considerable improvement over comparative algorithms in terms of accuracy and efficiency.

## 1. Introduction

Confronted with various malicious intrusions in today’s cyberspace, governments and enterprises have to deploy a series of network security devices to protect their information assets, such as firewalls, intrusion detection, and protection systems. These devices, however, often generate a large volume of low-level intrusion detection alerts from which it is difficult to obtain a full view of the ongoing cyber-attack [1]. The attack modeling technology is capable of transforming low-level intrusion detection alerts into high-level attack graphs via alert aggregation and correlation. Attack graphs enable network administrators to clearly understand the attack strategy that the intruders use to compromise the network such that they can implement a timely response to security threats.

The existing algorithms for attack modeling are generally classified into two categories—alert correlation and alert clustering [2]. The idea of alert correlation is to fuse and transform intrusion detection alerts into high-level meta-alerts and then build the attack model by correlating high-level meta-alerts based on similarity. The early alert-correlation-based modeling algorithms rely on manual input of prior data, rendering them impractical to generate attack models as the number of alerts increases. The problem can be solved by using the alert clustering approach, which automatically correlates intrusion detection alerts and stores them in a tree or graph structure. However, the generated attack graph may not be complete [3]. During recent years, some researchers have applied a process mining approach to network attack modeling.

Process mining, also known as workflow mining, aims to build a workflow model according to the execution of process instances recorded in the event log, which is mainly used for process analysis. Process mining aims to discover, monitor, and improve real processes by extracting knowledge from event logs readily available in today’s information systems [4,5,6]. Process mining can also effectively extract attack patterns from intrusion detection alerts [7]. However, in current works, researchers tend to build a large attack graph with numerous vertices and edges to describe all the cyber-attack behaviors. The problem is that such a graph is too complex to be analyzed by humans [8,9], which is essential for defending cyber-attacks in practice. Current models tend to build a large attack graph with numerous vertices and edges to describe all the cyber-attack behaviors. Such a graph is too complex to be analyzed by humans. Although a robust automatic checking technique can be implemented to analyze the attack graph. Manual analysis by the network security administrator is still an important means to ensure network system security in practice. Because an experienced network security administrator can effectively discover the potential danger latent in the attack graph. Furthermore, the network security administrator can carry out a directed and specific action to prevent attack from intruders. Although some researchers have proposed some methods to simplify or segment attack models, some problems remain such as low efficiency and information loss. Furthermore, regarding process mining, few related works consider employing a distributed architecture to process massive intrusion detection alerts. In previous studies, only some data pretreatment methods, such as a combination of duplicate alerts and assignment of priority to alerts, have been adopted to reduce the computational efforts [10,11,12]. Therefore, to improve the algorithm’s capability of processing massive security alert data, it is necessary to introduce a distributed architecture [13,14].

To address the aforementioned issues, we propose an attack modeling approach based on heuristic process mining and graph segmentation which is that performs well in practical situations and with results that are easy to understand [15]. We also design a distributed version of the attack modeling approach to address large-scale intrusion detection alerts. The main contribution of this study is summarized as follows:We employ the process-mining algorithm to generate an attack graph. By applying the process-mining algorithm, the security alerts are aggregated and the dependency relationship, short loop relationship, and long-distance dependency among them are analyzed. The chronological order of the related cyber-attack behaviors is also extracted and the exclusive OR(XOR in short)/AND relationship within each event are examined to obtain the logical relationship among cyber-attack behaviors and the attack graph is accordingly generated. Therefore, the process mining based algorithm can effectively extract the relation between security alerts and obtain deep insight into the attack strategies.To produce an attack graph more comprehensible to humans, we propose a graph segmentation algorithm for complex attack graphs. The proposed algorithm begins with a search for the branch points from which the subgraphs are split, followed by completion of the subgraphs according to their structure. An incremental update method for the subgraphs is also proposed to adapt to the dynamic change of the attack graph. The proposed algorithm reduces the complexity of the attack graph without ruining its structure facilitating manual analysis by the network security administrator.According to the standalone algorithms previously mentioned, a distributed attack graph generation algorithm based on MapReduce and a distributed cyber-attack graph segmentation method based on Spark GraphX are proposed to efficiently address massive security alerts.

## 2. Related Work

### 2.1. Generation of Attcak Graphs

During recent years, cyber-attack modeling has become a hot research topic. The aim of cyber-attack modeling is to discover the internal relationship among security alerts and generate an attack graph that can provide a global-oriented view of the attack strategy. Alert correlation, clustering, and process mining are the main methods employed in recent studies [16].

Alert correlation is among the major approaches applied to build cyber-attack models. The goal of alert correlation is to correlate low-level intrusion alerts and fuse them into high-level alerts (also known as meta-alerts or hyper-alerts) to be presented to network and security administrators. Lee et al. [17] proposed an alert correlation method based on alert feature similarity. The method filters and aggregates redundant alerts and then calculates the similarity between any two alerts based on a probabilistic correlation approach. Then, the similarity score is used to determine whether the alert pair can be aggregated to a meta-alert. Ning [18] proposed an alert correlation model based on analyzing intrusion prerequisites and consequences. The prerequisite of an intrusion is defined as the necessary condition for the intrusion to succeed while the consequence of an intrusion is regarded as the possible outcome. If an alert raised during the earlier stage is the prerequisite of a later alert, then the two alerts are correlated together. The method requires manual definition of the prerequisites and subsequences of several types of attacks making it unsuitable for large-scale network scenarios. In summary, although the aforementioned research provide some useful methods to correlate cyber-attack alerts, further analysis and abstraction is still required. More recent work relies on alert correlation techniques to extract cyber-attack patterns and generate attack graphs. Research in References [19,20,21] simulate the process of an attack by generating attack trees. The root of an attack tree is the network attacker’s target. Branches in the tree represent the sub-goals of the attack, which are denoted as interconnected nodes. The path between nodes represents different alternative paths that a network attacker can follow to achieve the goal. Ahmadinejad et al. [22] proposed a two-layer hybrid model to generate attack graphs. In the first layer, alerts are correlated based on casual relations, while in the second layer, a similarity-based algorithm correlates the alerts that are not correlated in the first layer. Spathoulas [23] proposed a three-phase system. First, alerts are aggregated according to feature similarity. Then, the aggregated alerts are correlated together to indicate potential threats. Although these methods are able to extract cyber-attack patterns, alerts are only merged at a low level and they are unable to obtain a high-level or global-oriented view of the attack strategy.

Alert clustering is another method widely used in cyber-attack modeling. Sadoddin et al. [24] proposed a real-time alert correlation algorithm based on incremental frequent structured pattern mining. The algorithm first presents a definition of virtual graph, whose nodes represent several real hosts of the network and edges a set of generalized alerts, to capture the common characteristics between different compositions of frequent patterns. Then, the algorithm generates frequent structured patterns from alerts based on their source and/or destination host connectivity. Furthermore, the signature of stable patterns is periodically mined from intrusion alerts via a frequent structured pattern mining algorithm designated as FSP_Growth. Lagzian et al. [25] proposed an alert correlation algorithm based on the frequent pattern of the graph structure. The algorithm aggregates alerts into a graph structure according to internet provider (IP) address and attack mode and applies the Bit-AssocRule algorithm to mine frequent patterns in the model graph. Ramaki et al. [26] proposed a real-time alert correlation framework based on stream mining to detect multi-step attack scenarios. First, the framework aggregate alerts into hyper-alerts and sorts them by their time tags. Then, it divides the hyper-alerts into different sequences, from which critical episodes are extracted to construct multi-level attack scenarios. Finally, the framework builds the attack trees to represent the exploited strategies of the multi-step attack. These clustering-based attack modeling algorithms can effectively mine intrusion data; however, a drawback is that the attack graphs generated by these methods are often far too complex.

Process mining can extract workflow information and build a process model based on the execution flow recorded in the logs without prior knowledge of the process patterns. Considering that the network intruder’s attack process is similar to a workflow, some researchers have applied the process mining method to attack modeling. Weijters et al. [15] proposed a heuristic process mining method that analyzes dependency relations between two attack events using a frequency-based metric. The algorithm performs well in analyzing event logs with noise. Alvarenga et al. [27] also proposed an attack modeling algorithm based on heuristic process mining. The method extracts intruders’ attack strategies from event logs grouped by attacker’s goal and attack time to generate an attack graph. This method can provide a full picture of the cyber-attack process. However, the generated attack graphs are too complicated for manual analysis.

### 2.2. Graph Segmentation

To describe the attack pattern extracted from the security alerts, the existing attack modeling approaches often build complex attack graphs. To improve their readability, it is necessary to split a complex graph into multiple subgraphs using graph segmentation. The graph segmentation methods can be approximately divided into two categories: approximate algorithms and exact algorithms. Approximate algorithms are mainly based on heuristic algorithms such as the linear programming [28], simulated annealing [29], genetic [30], ant colony [31], particle swarm optimization [32], spectral clustering [33,34,35], and K-L algorithms [36]. The heuristic algorithm can ensure segmentation of a large-scale graph at a reasonable time cost and computational power cost. The exact algorithm can obtain the exact segmentation result. Brunetta et al. [37] proposed a branch-and-cut algorithm based on linear relaxation and a separation strategy for the equicut problem on complete graphs. However, the time complexity is very high. Stefan et al. [38] presented a graph segmentation method for small graphs but it is not suitable for subgraph segmentation. Furthermore, the existing graph segmentation algorithms are mainly used in research fields such as complex network analysis, very-large-scale integration (VLSI) circuit design, and parallel computing. They are not applicable for segmentation of directed graphs whose nodes are closely linked with neighbors. This study proposes an effective graph segmentation algorithm based on the geometric properties of attack graphs.

## 3. Background Knowledge

Here we give a brief description of the process mining algorithm, and further details can be found in References [15,27]. Process mining algorithm can mine workflow without prior acknowledgement of process patterns. It receives as input an event log and returns as output a process model that is representative for the behavior observed in the event log. First, for process mining, each record in a log is considered as an event and each event corresponds to an action performed in the process. Second, each event in the process belongs to a process instance or case, which defines the scope of a process, that is, where a process starts and where it ends. As to attack modeling, each attacker composes a case and the attacker’s attack steps are the events of the case. Third, the occurring sequence of events is crucial for dependency mining because process mining relies on it to determine the ordering relationships between events. For example, let T be the case, W be the log on case T, A, B, C are the three events in the process of case T. If B is included in every directed path from event A to the ending event and no other events are on the path from A to B, then there is a dependency relationship between A and B. If A is followed by either B or C, then B and C have an AND relationship. If B and C do not execute simultaneously, then B and C have XOR relationship. During mining, it is possible that the same event is executed multiple times. This is called a short loop in the process. Short loops can be addressed by the use of dependency/frequency table (D/F table) and dependency score. The D/F-table indicates the frequency of ordering relationships occurrence, for example, number of times one event is directly followed by another event, while the dependency score represents the confidence that there is a dependency relation between two events.

## 4. Attack Modeling

### 4.1. Framework of Proposed Algorithm

The proposed attack-modeling algorithm attempts to aggregate low-level, less readable cyber alerts into cyber attack graphs. As shown in Figure 1, the framework of the algorithm consists of three modules which including two part: Generation of initial attack graph and segmentation of attack graph.

**Preprocess of security alerts**: The raw alert data are cleaned to remove noise and duplicates and then grouped and aggregated before converting to the required format for the process mining algorithm.

**Generation of initial attack graphs**: The heuristic process mining algorithm is executed to extract an attack pattern from security alerts and build an initial attack graph.

**Segmentation of complex attack graphs**: The graph segmentation algorithm is performed to segment complex attack graphs into simple and more readable subgraphs while preserving the attack information in the original attack graphs.

### 4.2. Initial Attack Graph Generation

First, the heuristic process mining algorithm is used to mine attack graphs from security alerts. As is shown in Figure 2: In the first step, alert logs data will be aggregated, to gather the alert with common features. For example, the attack alert from the same destination IP in a period will be gather to one group. And then the log will be transform to XES (eXtensible Event Stream) format that will be adopted to process mining program. And then, the miner will calculate the dependency score according to the frequency behavior from input data of which value is from −1 to 1. The dependency score is a value that is used to validate if the relationship between two attack behavior exists. A dependency/frequency table will be constructed to detect the short loops. Meanwhile, the algorithm will calculate the frequency of relationship between behaviors to detect the XOR/AND relation and long distance dependency. After that an attack graph with complete relationship information in attack behaviors would be generated. The principle and the effectivity of Heuristic Mining algorithm can be found in References [15,27,39]. It is out of scope to discuss much in detail because this study focuses on simplify the result attack graph to be suitable.

### 4.3. Attack Graph Segmentation

Manual analysis of cyber-attacks by network security administrators is an important means to ensure network system security in practice [26]. However, attack graphs generated by state-of-the-art algorithms are extremely complex for network security administrators to understand. Taking the heuristic process mining based attack graph modeling algorithm as an example, when process mining is performed on attack behaviors, the algorithm first identifies attack behaviors and then marks them as vertices and their orders as edges in the attack graph. Because of the massive number of security alerts and the complex dependencies and relationships among them, the number of vertices and edges are enormous. Consequently, the attack graph will be too complex to be manually analyzed. Obviously, such a large and complex attack graph is not intuitive for network administrators to analyze the network security status. To address this issue, we design a heuristic graph segmentation algorithm to segment the complex attack graph into multiple attack subgraphs, which can more clearly and concisely represent the intruder’s attack steps.

#### 4.3.1. Problem Definition

The attack graph $G_{t}=\left(V_{t},E_{t}\right)$ illustrates the relationship among attack behaviors at time *t*, where $V_{t}$ denotes a set of vertices in the attack graph, representing the intruder’s attack behaviors, and $E_{t}$ denotes a set of edges in the attack graph, representing the logical relationship among the intruder’s attack behaviors. The problem of attack graph segmentation is defined as follows:

**Definition** **1.**
*Attack graph segmentation. Given an attack graph $G_{t}$ at time t, find the graph segmentation $P_{t}$ of the attack graph $G_{t}$, and $P_{t}=\{{G_{t}}^{\left(1\right)},{G_{t}}^{\left(2\right)},\ldots ,{G_{t}}^{\left(n\right)}\}$ where ${G_{t}}^{\left(i\right)}$ represents the ith subgraph of the attack graph $G_{t}$.*


#### 4.3.2. Algorithm Overview

The basic idea of the proposed algorithm is as follows: first, traverse the initial attack graph to find the branch point and then proceed from there to visit other vertices and edges before finally saving the traversed vertices and edges as a new attack graph. The process does not change the original attack graph structure. The algorithm consists of six steps as follows:

*Step 1*: Calculate the complexity of the attack graph. If the attack graph is a complex graph, place it into the queue $Q_{s}$.

*Step 2*: Traverse through the queue $Q_{s}$, for each graph *G* in the queue $Q_{s}$ and look for the branch point $V_{split}$ from top to bottom.

*Step 3*: From the branch point *v*, first remove independent subgraphs from the attack graph and then add the directed edges of which *v* is the starting point to the queue $Q_{e}$.

*Step 4*: Traverse through the queue $Q_{e}$, perform a depth-first search from each directed edge *e*, generate successor subgraphs of *e*, and mark *v* as the starting point of the successor subgraphs. For each successor subgraph, calculate its complexity. If it is still a complex subgraph, add the subgraph to the queue $Q_{s}$ for further segmentation; otherwise, the successor subgraph is added to the queue $Q_{output}$ that stores subgraphs that have finished segmentation.

*Step 5*: Repeat Steps 2–4 until the queue $Q_{s}$ is empty.

*Step 6*: Traverse through the queue $Q_{output}$, for each subgraph in the queue $Q_{output}$, complete the branch information, and then output the resulting subgraphs that consist of the attack graph *G*. More details of the proposed algorithm are presented in the following sections.

#### 4.3.3. Evaluating Attack Graph Complexity

The purpose of attack graph segmentation is to reduce the visual complexity of an attack graph, facilitating understanding by network security administrators. Therefore, we first define the rules to evaluate the complexity of an attack graph to determine whether to segment the attack graph or not.

*Rule 1*: Given an attack graph $G_{t}=\left(V_{t},E_{t}\right)$, where *V* denotes the set of vertices and *E* denotes the set of edges, if the number of vertices in *G* satisfies $\left|v\right|<N_{1}$, *G* is a simple attack graph and no further segmentation is required.

*Rule 2*: Given an attack graph $G_{t}=\left(V_{t},E_{t}\right)$, if the number of vertices in *G* satisfies $\left|v\right|>N_{2}$, the graph *G* is a complex attack graph and requires further segmentation.

*Rule 3*: Given an attack graph $G_{t}=\left(V_{t},E_{t}\right)$, if the number of vertices in an attack graph satisfies $N_{1}<\left|v\right|<N_{2}$, then whether *G* is a complex attack graph or a simple attack graph is determined by the $Simplicity\left(G\right)$. If the $Simplicity\left(G\right)$ is less than the threshold γ, the graph *G* is a complex attack graph, and requires further segmentation.
(1)$$Simplicity\left(G\right)=\frac{\left|V\right|}{\left|E\right|}.$$

Based on the aforementioned analysis, the final criterion is shown in Equation (2), where $N_{1}=15$ and $N_{2}=31$ [16], and the threshold value of γ is 0.302 according to an expert’s experience.
(2)$$ComplexDegree\left(G\right)=\left\{\begin{array}{c} 0\;\;\;\;E<N_{1}\;\\ 1\;\;\;\;N_{1}<E<N_{2}\\ \;\;\;\;\;\;\;\;\;\&Simplicity(G)<\gamma \\ 0\;\;\;\;N_{1}<E<N_{2}\\ \;\;\;\;\;\;\;\;\;\&Simplicity(G)⩾\gamma \\ 1\;\;\;\;E>N_{2} \end{array}.\right.$$

#### 4.3.4. Segmentation of Independent Subgraphs

**Definition** **2.**
*Successor subgraph. Given a vertex v in an attack graph G, starting from a directed edge e that ends with v, the reachable vertices and edges compose a successor subgraph of v, denoted as ${G_{succ}}_{\left(v,e\right)}$.*


As shown in Figure 3, the part in the solid line composes the successor subgraph of vertex *v*. In an attack graph *G*, vertex *v* represents an intruder attack behavior and the successor subgraph of *v* indicates the possible subsequent intruder attack behaviors after the attack behavior represented by *v*.

**Definition** **3.**
*Branch point. Given a vertex v in an attack graph G, if the out-degree of v is larger than 1, v is a branch point of attack graph G.*


A branch point *v* in the attack graph indicates that the intruder has more than one possible attack scheme after the attack behavior represented by *v*. By conducting segmentation from the branch point, different attack schemes can be separated as independent subgraphs such that the administrator can clearly understand each attack scheme. The proposed algorithm searches for the branch point *v* in the attack graph from top to bottom and separates the subgraphs.

**Definition** **4.**
*Independent subgraph. Given a successor subgraph of vertex v, for each vertex in this success subgraph, all the directed edges that end with this vertex should belong to the successor subgraph. Then this subgraph is an independent subgraph with v as the starting vertex.*


As shown in Figure 4, the subgraph in the solid line represents an independent subgraph of vertex *v*. First, the directed edges with *v* as the end vertex are obtained from the attack graph. Then, starting from the directed edges, the successor subgraphs of *v* are obtained by traversing through the graph. If the subgraph is an independent subgraph with *v* as the starting vertex, the subgraph will be separated from the original graph and placed into the queue $Q_{s}$ where it will be examined later to decide whether further segmentation is required.

In an independent subgraph, all the attack behaviors denoted by this subgraph are and only are the follow-up steps of the attack behavior represented by the branch points. When the network security administrator discovers a subsequent attack indicated by the independent subgraph, the previous attack behavior can be confirmed. Note that it is not necessary to complete the independent subgraph if it is a successor subgraph of a branch point. Therefore, the proposed algorithm will first separate the independent subgraph from the attack graph to improve the computational efficiency of the subsequent segmentation operation without affecting the segmentation result of the remainder of the attack graph.

#### 4.3.5. Segmentation of the Remaining Subgraphs

After the independent subgraphs are segmented from the initial attack graph, the proposed algorithm needs to process the remaining subgraphs that start with branch point *v*. Starting from the directed edges with *v* as the end point, the proposed algorithm will traverse and segment each successor subgraph that starts with the branch point *v*. The processing steps to segment the remaining subgraphs are as follows:

*Step 1*: Traverses the queue $Q_{e}$, for each directed edge *e* in the queue $Q_{e}$, and labels the successor vertices and their incident edges by breadth-first traversal. All the reachable vertices and edges from *e* constitute a candidate subgraph.

*Step 2*: Removes the long-distance cycle in the candidate subgraph and generates the final subgraph.

Take the initial attack graph in Figure 5a as an example; the subgraphs that start with the branch point *v* are segmented from the initial attack graph by the following steps shown in Figure 5b–d.

First, the proposed algorithm performs a depth-first traversal of the initial attack graph from the directed edge $e_{1}$ and visits all the vertices and edges that can be reached from $e_{1}$. The discovered vertices and edges together with the starting vertex *v* compose the subgraph $G_{1}$, as shown in Figure 5b. Likewise, the vertices and edges traversed from $e_{2}$ and $e_{3}$, together with the starting vertex *v*, compose the subgraphs $G_{2}$ and $G_{3}$, as shown in Figure 5c,d, respectively.

The subgraph extracted from the attack graph represents the attack strategy in a certain attack stage of the entire cyber-attack. The branch in the entire attack graph indicates that the intruder may execute different attack strategies. In a subgraph, a vertex corresponds to a certain attack behavior that occurred during the cyber-attack process and the vertices in the same subgraph are possible behaviors before and after the certain attack behavior. The vertices that do not belong to the same subgraphs correspond to attack behaviors that will not be the precedent or subsequent attack behaviors of the certain attack behavior. Therefore, network security administrators can only focus on one subgraph to investigate the dependency and logical relationship of the attack behavior with its predecessor and successor.

#### 4.3.6. Remove Long-Distance Cycle

**Definition** **5.**
*Long-distance cycle. A long-distance cycle is a path that begins and ends at the same vertex and it may pass through a vertex more than once. Long-distance cycles involve long-distance dependency resulting from the processing mining algorithm. As shown in Figure 6a, which is a part of the subgraph $G_{1}$ previously mentioned, the algorithm travels back to the starting vertex v after it traverses along the path indicated by the solid line and the traversal path constitutes a long-distance cycle. The presence of a long-distance cycle may result in many unnecessary iterations, increasing the complexity of the proposed graph segmentation algorithm. Hence, two different strategies are proposed to remove the long-distance cycle according to the circumstances.*


In the first case, from vertex *v*, the proposed algorithm traverse downwards until it reaches vertex $v_{1}$. If vertex $v_{1}$ does not belong to subgraph $G_{1}$ and vertex $v_{1}$ and *v* are not the same vertex, then $v_{1}$ is considered to be a vertex beyond subgraph $G_{1}$. The algorithm will move back to the previously visited vertex *t* and then remove vertex $v_{1}$ and the directed edge from $v_{1}$ to *v* from the candidate subgraph $G_{1}$. The vertices and the edges indicated by the solid lines correspond to the final subgraph $G_{1}$ after removing the long-distance cycle, as shown in Figure 6b.

In the second case, if the proposed algorithm discovers a directed edge *e* points to vertex *v*, the algorithm will move back to the previously visited vertex *t* and then the directed edge ending with vertex *v* is added to the subgraph $G_{1}$. The vertices and the edges indicated by the solid lines correspond to the final subgraph $G_{1}$ after removing the long-distance cycle, as shown in Figure 6c.

#### 4.3.7. Completing Generated Subgraphs

After segmentation, the algorithm needs to complete the resulting subgraphs to facilitate further analysis by network security administrators. The predecessors of the vertices within the subgraph need to be completed. These predecessors represent the precedent attack steps of an intrusion; however, some predecessors will not be included in the corresponding subgraph because the traversal algorithm moves forward along the directed edge and is unable to visit the predecessors of some vertices. To facilitate administrators understanding the attack steps the intruder takes before and after execution of an attack behavior, these predecessor vertices need to be included in the subgraph. In addition, these predecessor vertices also exist in some other subgraphs at the same time. Therefore, completing the information of these vertices can correlate different subgraphs and help the network security administrator understand the logical relationship among different subgraphs. The steps to complete a subgraph are as follows:

*Step 1*: Traverse each branch vertex *v* in the subgraph to be completed. Find all the directed edges that end with vertex *v* in the original Graph *G* denoted as $E_{v}$.

*Step 2*: Traverse the edge set $E_{v}$; for each directed edge *e* in $E_{v}$, if *e* does not belong to the subgraph to be completed, add *e* to the subgraph.

Taking the subgraph $G_{1}$ shown in Figure 5b as example, the algorithm completes subgraph $G_{1}$ and the newly added vertices and edges are marked in bold lines, as shown in Figure 7.

#### 4.3.8. Algorithm Description

The pseudo code of the proposed algorithm is shown in Algorithm 1.

### 4.4. Analyzing Algorithm Complexity

The time complexity of the process mining algorithm applied to generate the initial attack graph is related to the number of attack steps *k* to be discovered and denoted as $O(k^{2})$ [39].

We analyze the time complexity of the proposed graph segmentation algorithm by calculating the time complexity of each step.

*Step 1*: The time complexity to traverse an attack graph is O(m), where *m* is the number of vertices in the attack graph.

*Step 2*: The time complexity to search for branch points is also O(m).

*Step 3*: The time complexity to segment independent subgraphs is O(xn), where x represents the number of branch points and n the number of edges.

*Step 4*: The time complexity to segment the remainder of the attack graph is also O(xn).

*Step 5*: The time complexity to complete each subgraph is related to the number of vertices that requires completion of information and is O(nm).

Based on the aforementioned analysis, the time complexity of the proposed graph segmentation algorithm is O(n(m+x)) and the running time is closely related to the structure of the initial attack graph.

Regarding the space complexity, the memory required depends on the number of vertices m and the number of edges *n* such that the space complexity is O(m+n).
 **Algorithm 1** Complex Graph Segmentation Algorithm1:functionAttackGraphGenerater(AttackGraphG)2:**if** 
ComplexDegree(G) 
**then**3: 
$Q_{s}.push\left(G\right)$
4: **while** 
$\left(Q_{s}\right)$ 
**do**5:  $G<-Q_{s}.pop\left(G\right)$6:  /* Looking for branch points */7:  $V_{Split}<-SearchSplitVertex\left(G\right)$8:  /*Judge independent subgraph */9:  **if** 
$IsDependentGraph(V_{Split})$ 
**then**10:   /* Independent subgraph segmentation */11:   $DependGraphGenerater(V_{Split})$12:  
**end if**
13:  /* saves the edge to be traversed */14:  $Q_{e}.push\left(getSuccEdge\left(V_{Split}\right)\right)$15:  **while** 
$\left(Q_{e}\right)$ 
**do**16:   /* Looking for successor subgraphs */17:   $G_{e}=getSuccGraph\left(Q_{e}.pop\left(\right)\right)$18:   **if** 
$ComplexDegree(G_{e})$ 
**then**19:    $Q_{s}.push\left(G_{e}\right)$20:    /* saves the output graph */21:   
**else**
22:    $Q_{Output}.push\left(G_{e}\right)$23:   
**end if**
24:  **end while**
25: **end while**
26: **while** 
$\left(Q_{Output}\right)$ 
**do**
27:  /* subgraph Completion */28:  $G=InfoComplete(Q_{Output}.pop\left(\right))$29:  Output(G)30: **end while**
31:**else**32: Output(G)33:**end if**

## 5. Distributed Network Attack Modeling

Recently, distributed computing models such as Hadoop MapReduce or Spark have been widely used for processing large-scale data that cannot be processed in a single machine. When the number of alert data reaches a certain scale, the aforementioned standalone attack modeling algorithm will not be able to address the massive alert data in an efficient and timely manner. Therefore, this section discussed how to adapt the standalone attack modeling algorithm to the distributed framework.

The framework of the distributed attack modeling algorithm is shown in Figure 8. The Hadoop Distributed File System (HDFS) is used to store the raw security alerts, initial attack graph generated by the process mining algorithm, and subgraphs obtained by the proposed attack graph segmentation algorithm. The HADOOP MapReduce is used to generate the initial attack graph and Spark GraphX is used for segmentation of the initial attack graphs in a distributed manner. The Yet Another Resource Negotiator (YARN) is responsible for the unified management of MapReduce and SparkGraphX and used to assign jobs to different computer nodes.

### 5.1. Distributed Initial Attack Graph Generation

The process of the distributed process mining algorithm is shown in Figure 9. The algorithm consists of four modules: SecurityCaseRelationCreation, AttackRelationComputation, AttackClassification, and CausalMatrixGeneration, of which detail as shown in Algorithm 2. The principle of the four modules are shown as figure: XES logs are divided into several part and then mined in each node. After the relationship between every two steps are mined, they will be combine the matrix into a complete attack graph.

SecurityCaseRelationCreation: The first module SecurityCaseRelationCreation aims to extract the logical relationships among security alerts. This map task termed SecurityXesLogCaseMapper reads raw security alerts (XESlog) from HDFS and identifies their source IP, signature, and other fields as required. It then splits the security alerts into multiple sublogs using source IP as the key. Each sublog is used as a case for processing mining. Furthermore, in the map task of distributed processing mining, the signature is set as an event, that is, an attack behavior in the attack graph, and the timestamp is set as a key for secondary sorting. The result is then output to the reduce task termed AttackCaseReducer where the relationship between attack events in the same case is computed and the ordering of attack steps is extracted. As shown in Figure 10, for instance, *A* and *B* are two events in the same case, where B occurs after A. We can use AttackCaseReducer to determine their order.
**Algorithm 2** AttackGraphMiningAlgorithm1://CaseRelationCreation2:SecurityXesLogCaseMapper(key:RowNumber,value:Event,Timestamp)3:Output(key:case timestamp,value:Event)4:AttackCaseReducer()5:Output(key:relation(event,event),value:Relation metrics)6://RelationComputation7:RelationReducer(key:Relation(event,event),value:List relationMetrice)8:Output(RelationMetrice(Aggregated))9://EventClassification10:ScoredEventMapper(key:Relation(Event,Event), value:RelationMetrics(Aggregatged))11:Output(key:singleton, value:ScoredEvent(input,Output,relations))12:ScoredEventReducer()13:Output(key:Event, value:ScoredEvent(input,Output,relations))14://CausalMatrixGeneration15:CausalMatrixMapper(key:Event, value:ScoredEvent(input,Output,relations))16:Output(key:singleton, value:ScoredEvent(input,Output,relations))17:CausalMatrixReducer(AttackGragh)

AttackRelationComputation: The second module AttackRelationComputation only performs the reduction task RelationReducer. The output of AttackCaseReducer is taken as input of the reduction task RelationReducer which aggregates the relationships among cyber-attack behaviors and summarizes the different types of relationships such as order, branch, and short-loop relationship. Finally, RelationReducer establishes a relation matrix to store all types of aggregated relationships, as shown in Figure 11a.

AttackClassification: The ScoredEventMapper further partitions the data according to the left value in the relation pair (for example, the predecessor in the order relationship). Thereafter, the ScoredEventReducer outputs the predecessor vertex, successor vertex, and the edge between them (i.e., the relationship between the two events) as results.

CausalMatrixGeneration: The CausalMatrixMapper finds the predecessor and successor vertices based on the vertex corresponding to the key and combines them into a relational matrix. The CausalMatrixReducer aggregates the output from each computing node to compose a complete attack graph, as shown in Figure 11b.

### 5.2. Distributed Attack Graph Segmentation

Apache Spark’s GraphX is quite suitable for efficiently processing graph data and can be used for distributed attack graph segmentation. Spark uses Resilient Distributed Datasets (RDDs) as data abstraction. Therefore, to perform distributed attack graph segmentation under SPARK GraphX, the graph data need to be converted to three RDDs: VertexTable, EdgeTable, and RoutingTable, where VertexTable stores information of vertices, EdgeTable stores information of edges, and RoutingTable stores the locations of vertices in a cluster.

Whether to perform distributed attack graph segmentation depends on the complexity of the attack graph. If the attack graph is a simple graph that only has tens of vertices, considering the communication cost between multiple processing nodes, it would be more feasible to segment the attack graph in a single node. In contrast, if the attack graph has a large number of vertices, the attack graph should be decomposed into several partitions by calling Graph.partitionBy and then distributing to multiple processing nodes for segmentation.

## 6. Experiment and Analysis

### 6.1. Experimental Setting and Dataset

#### 6.1.1. Experimental Environment and Dataset

In this section, we evaluate the distributed attack graph modeling algorithm by conducting extensive experiments. We implement the algorithm by Java, using the Apache Hadoop and Apache Spark distributed framework. The experimental settings are shown in Table 1.

We use the security alerts generated by intrusion detection system/intrusion prevention system (IDS/IPS) devices installed in a European Internet service provider between March and August 2016 as the experimental dataset. Each security alert in the experimental dataset records a cyber-attack behavior. However, the intruders’ attack strategy is not described. The main fields of a security alert are listed in Table 2. Note that, to facilitate researchers validating their models, the security alerts raised between June and July are classified by common attack modes.

The size of the experimental data exceeds 11 GB. For convenience, we select the security alerts raised during a week in July to validate the proposed algorithm precision. After preprocessing the raw security alerts, we group the data by day and cluster the data into different groups as input cases with a source IP field as the key. In the single-node experiment, we chose ProM framework to generate the initial attack graph, which is among the most popular process mining tools currently available. The security alerts are converted into the eXtensible Event Stream (XES) format to be imported to ProM based on the dates the alerts were raised.

#### 6.1.2. Comparing Algorithms and Evaluation Metrics

During the experiments, we chose the Alpha [40] algorithm-based attack graph mining algorithm and the attack model discovery algorithm (MDA) proposed by Reference [16] for comparison. The precision rate, recall rate, and the overall index F1−score are used to evaluate the performance of the proposed algorithm as follows:(3)$$Precision\;=\;\frac{TP}{TP+FP}$$
(4)$$recall=\frac{TP}{TP+FN}$$
(5)$$F1-score=2\cdot \frac{Precision\cdot Recall}{Precision+Recall},$$
where TP represents the attack sequence that appears in both the generated attack graph and the reference information in the data set or the control group, indicating that the attack sequence is effectively extracted. FN represents the attack sequence that does not appear in the generated attack graph, meaning that the algorithm fails to extract the attack sequence.

### 6.2. Experiment Analysis

#### 6.2.1. Analysis of Attack Graph Generation

The security alerts raised from July 26 to July 30 are clustered into five groups; the results are shown in Figure 12.

As shown in Figure 12, the recall rate of the heuristic process mining algorithm (HPM) reaches 91.3%, the precision rate reaches 85.7%, and the F1-score reaches 88.8%, which are better than the other two compared algorithms. The experiment results prove that the HPM can cover most of the attack sequence that occurred. Thus, the HPM can effectively extract the intruder’s attack strategy and possible attack steps from the security alerts because it can better eliminate the influence of noise during the mining process.

#### 6.2.2. Analysis of Attack Graph Segmentation

Although the raw security alerts are grouped by day, the initial attack graph generated is still very complex because of the massive volume of security alerts. For example, on 31 July 2016, 323,120 security alerts were recorded and grouped into 4367 cases with source IP as the index and 79 events on average. The output is a complex attack graph consisting of 34 vertices and 109 edges. Therefore, we analyzed the performance of the attack graph segmentation as described in this section. Taking the initial attack graph generated using the data of the last week of July as an example, the initial attack graph is partitioned into 14 subgraphs. The number of vertices and edges of each subgraph are shown in Table 3, including 13 simple subgraphs and one complex subgraph (No. 11). Tracing back, it was found that the No. 11 subgraph changed from a simple subgraph to a complex subgraph because of subgraph completion.

Here, we analyze the No. 11 subgraph to verify the validity of the segmentation results. As shown in Figure 13, it can be seen from the No. 11 attack subgraph that the intruder who performed the sshScan scan attack will normally complete four types of attack behaviors during the next step: Impossible Flags (using the flag of the invalid Transmission Control Protocol (TCP) header, a denial-of-service (DOS) attack), Nmap scanning, malicious Server Message Block (SMB) probes, and malicious PHP programs. The first three types of attacks may be subsequent attacks of one another. After using the SMB probe, the attacker will use the Windows system vulnerability for the Microsoft SQL Hello Buffer Overflow attack (an attack that exploits MS SQL vulnerabilities) or cause a Windows Plug and Play exception request. However, if the intruder uses a PHP malicious program after sshScan scan, the intruder will use the PHP code injection to execute a further attack. From the aforementioned example, it is obvious that the attack subgraphs’ output by the proposed algorithm can clearly reflect the intruder’s attack pattern.

#### 6.2.3. Time Performance Analysis

##### Time Performance of HPM

We analyze the time performance of the standalone version and distributed version of the proposed HPM algorithm and Alpha algorithm. The distributed version runs on a Hadoop cluster of four nodes. We chose the security alerts raised during two weeks of July and the amount of security alerts was 11.4, 8, 4, and 1 GB, respectively. The time performance of the four algorithms is shown in Figure 14.

From Figure 14, we can see that the advantages of the distributed mining algorithms become obvious with the increase in the log size and the HPMoMR runs slightly longer than the distributed Alpha algorithm. However, both are significantly better than the stand-alone version of the HPM and the Alpha algorithm. The runtime of the distributed Alpha algorithm is less than that of the HPMoMR because the Alpha algorithm could not mine the repeated tasks but the calculation amount is instead much smaller. The main process of the Alpha algorithm is simpler and faster compared to that of the HPM algorithm; however, the attack graph generated by the HPM algorithm is better than that by the Alpha algorithm.

Next, we analyze the impact of the number of nodes on the runtime of the compared algorithms with a log size of 11 GB. The experimental result is shown in Figure 15 and Figure 16. With two cluster nodes, the HPMoMR runtime is reduced by 39% compared to that of the standalone version. With four cluster nodes, the runtime is reduced by 62% because the original data can be simultaneously processed in multiple nodes, which effectively alleviates the Input/Output bottleneck. However, with eight nodes, the runtime is only reduced by 67% mainly because the I/O bottleneck can be effectively addressed in the case of four nodes. In addition, in the MAP task, the number of partitions increases as the number of nodes increases and the communication cost between nodes increases. Therefore, the time performance does not appreciably improve as the number of nodes increases from 4 to 8.

##### Time Performance of Graph Segmentation

The runtime of the distributed attack graph segmentation algorithm is analyzed and compared. The initial attack graph is used as the input of Spark GraphX. The number of vertices of the initial attack graph is 276 and the number of edges is 1053. The runtime of the distributed graph segmentation algorithm is significantly better than that of the standalone version. The time performance of the distributed algorithm does not appreciably change as the number of cluster nodes increase from 4 to 8 because of the communication cost between multiple nodes. In addition, when segmenting a graph with hundreds of vertices, Spark can still efficiently output results.

## 7. Conclusions

We propose an attack modeling approach that uses a heuristic process mining technique to effectively extract intruder attack strategy. The main concept of this approach is to exploit the similarity between the intruder attack behavior and the workflow characteristics and use the process mining to mine intrusion alerts. Furthermore, to reduce the attack graph complexity, we propose a segmentation algorithm for complex attack graphs. Under the premise of preserving the basic structure of the attack model, the algorithm successfully divides the complex attack graph into multiple low-complexity attack subgraphs, significantly increasing the graph readability and enabling network security administrators to obtain more detailed attack information to make appropriate decisions.

Based on the aforementioned method, we propose an attack graph generation algorithm based on Hadoop MapReduce and an attack graph segmentation algorithm based on Spark Graphx, which significantly improves the attack graph mining efficiency.

## Figures and Tables

**Figure 1 entropy-22-01026-f001:**
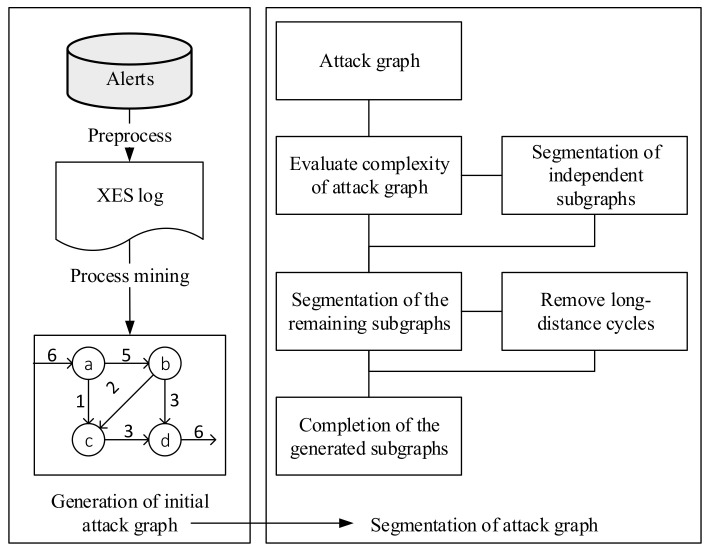
Framework of the proposed attack modeling algorithm.

**Figure 2 entropy-22-01026-f002:**
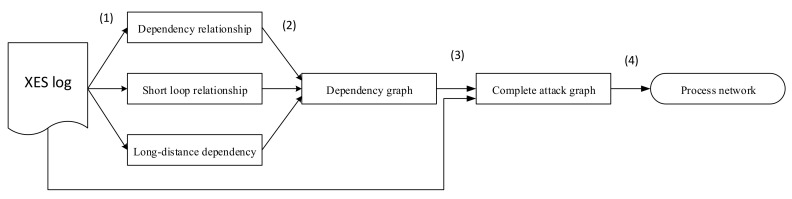
Flowchart of heuristic process mining.

**Figure 3 entropy-22-01026-f003:**
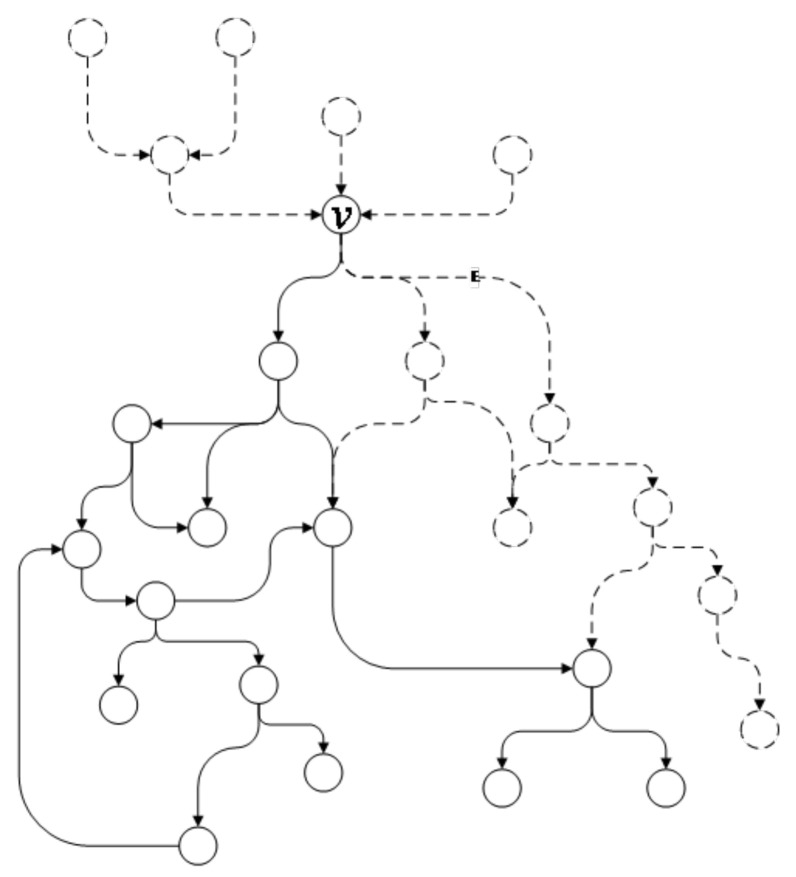
Subgraph of vertex *v*.

**Figure 4 entropy-22-01026-f004:**
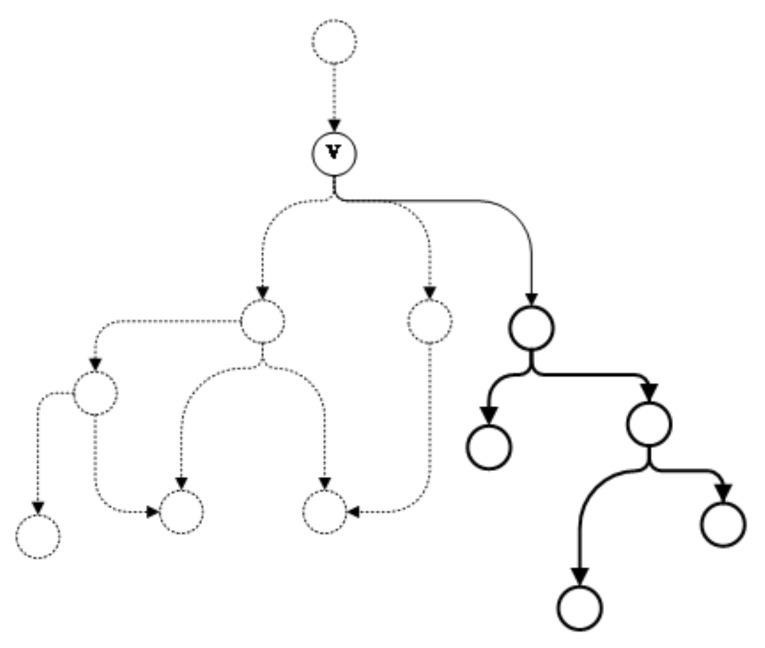
Independent subgraph.

**Figure 5 entropy-22-01026-f005:**
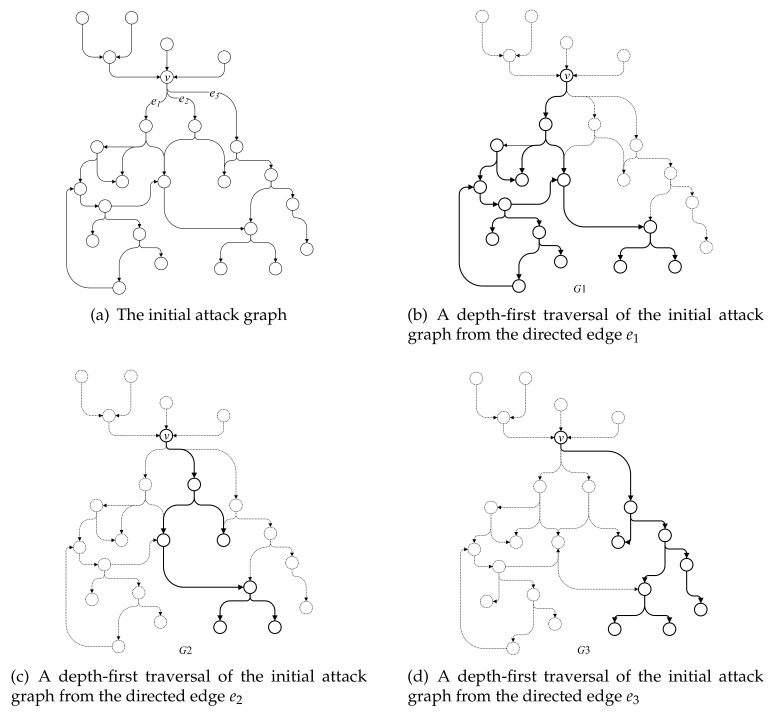
Successor graph split into three subgraphs with *v* as the starting vertex.

**Figure 6 entropy-22-01026-f006:**
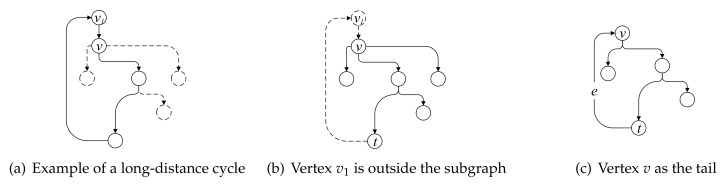
Long-distance cycle.

**Figure 7 entropy-22-01026-f007:**
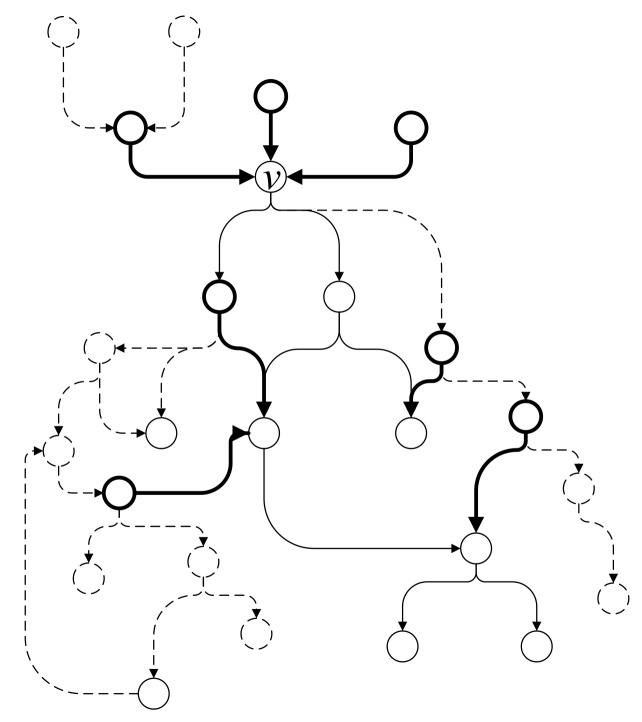
Example of subgraph completion.

**Figure 8 entropy-22-01026-f008:**
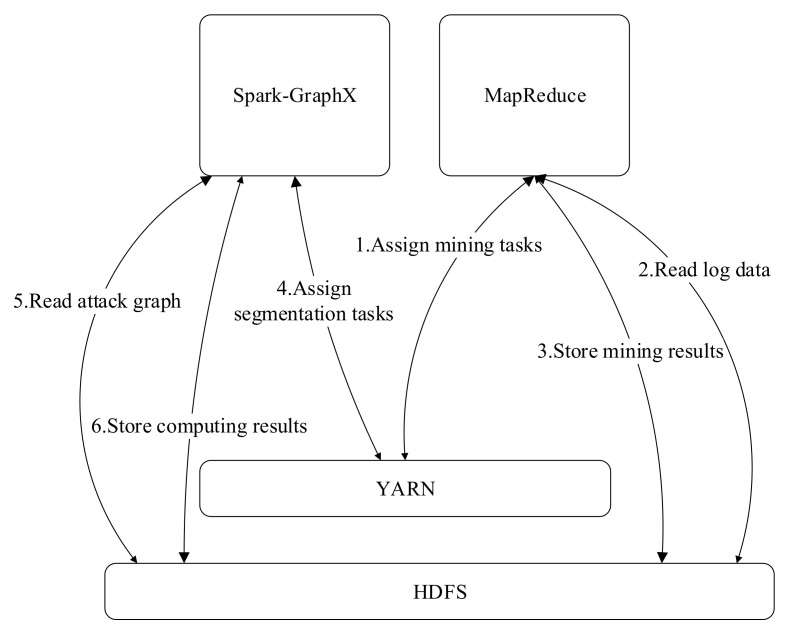
Working process of the distributed attack graph generation system.

**Figure 9 entropy-22-01026-f009:**
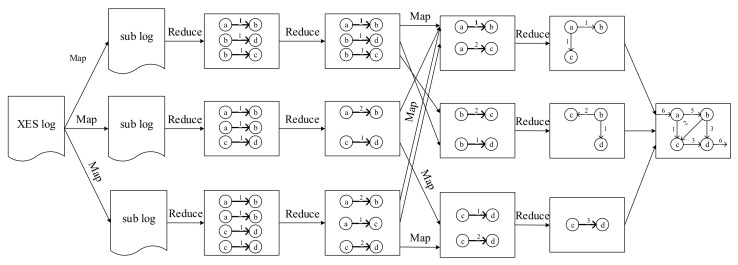
Process of the distributed process mining algorithm.

**Figure 10 entropy-22-01026-f010:**
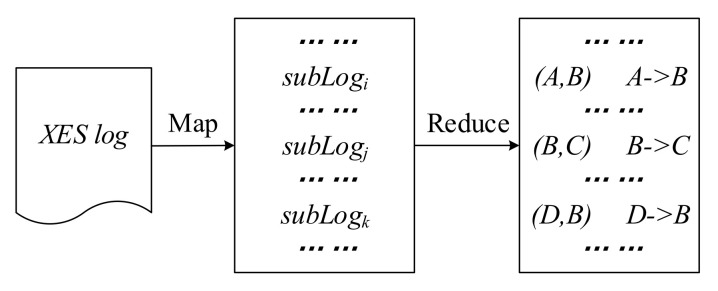
SecurityCaseRelationCreation.

**Figure 11 entropy-22-01026-f011:**
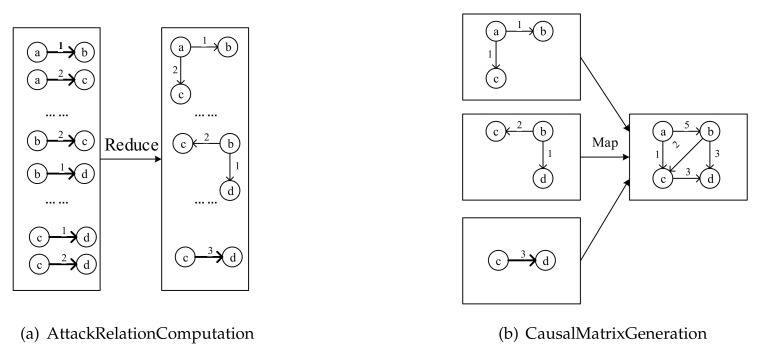
AttackRelationComputation and CausalMatrixGeneration.

**Figure 12 entropy-22-01026-f012:**
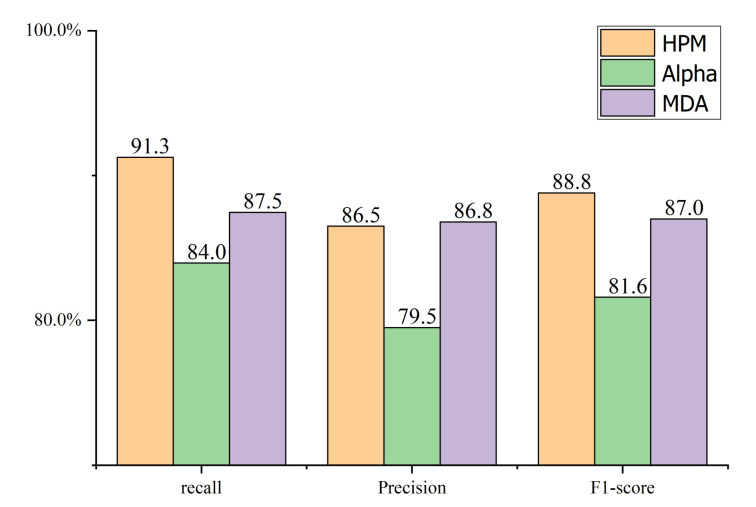
Comparison of recall, precision and F1-score.

**Figure 13 entropy-22-01026-f013:**
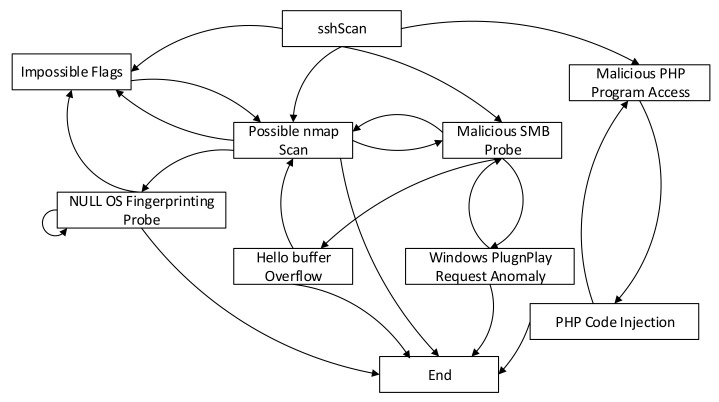
Attack Steps of the No.11 attack subgraph.

**Figure 14 entropy-22-01026-f014:**
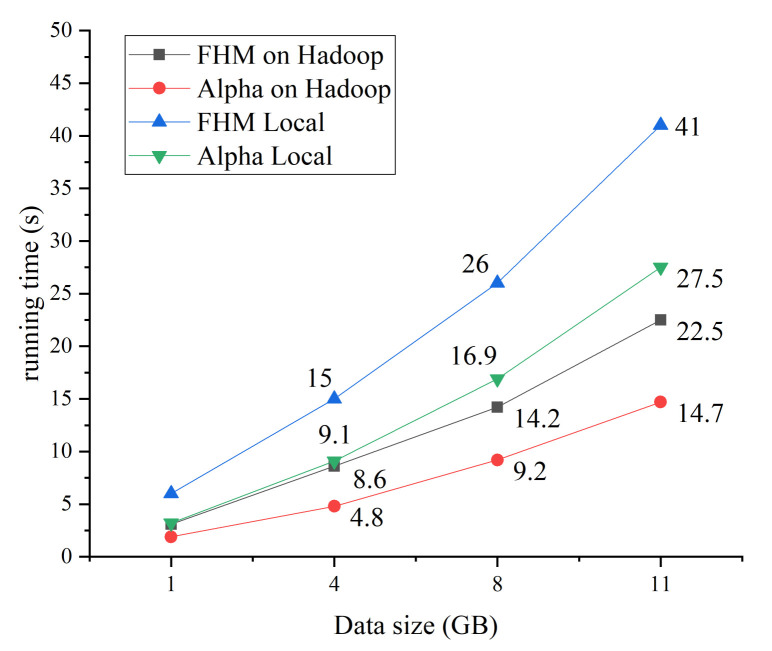
Impact of data size on runtime of attack graph modeling algorithms.

**Figure 15 entropy-22-01026-f015:**
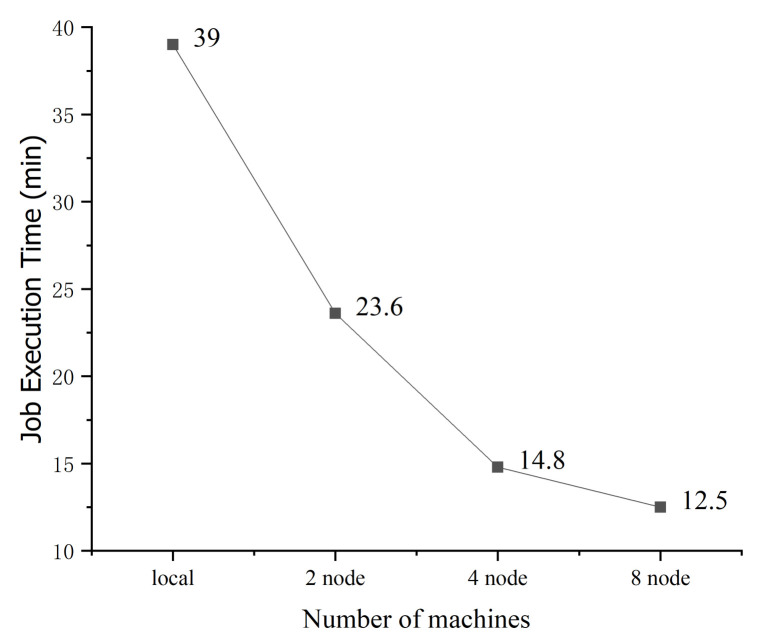
Impact of the number of nodes on the runtime of the attack graph generation algorithm.

**Figure 16 entropy-22-01026-f016:**
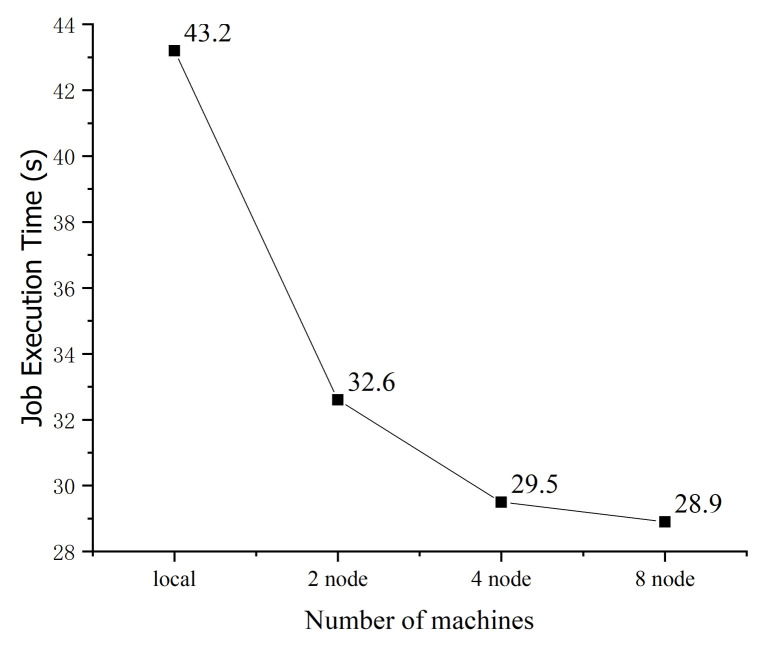
Runtime of the segmentation algorithm considering different numbers of nodes.

**Table 1 entropy-22-01026-t001:** Experimental environment configuration.

Parameter	Value
Processor	4
OS	Ubuntu 14.04
Hadoop	2.6.0
Spark	2.2.0
Number of nodes in a cluster	4

**Table 2 entropy-22-01026-t002:** Field description.

Field	Meaning	Description
Timestamp	Timestamp	Specifies the time when alerts occur; used as a basis for grouping
SourceIp	Source IP	Specifies the IP address initiating an attack
SourcePort	Source port	Specifies the port number of the attack source
DestIP	Destination IP	Specifies the IP address of the target host; used as a basis for grouping
DestPort	Destination port	Specifies the port number of the target host
Protocol	Protocol	Specifies the protocol of the traffic
Signature	Signature	Specifies the attack information; used as the event input for process mining
LogID	Log ID	Specifies the primary key of a log; the unique identifier of an alert

**Table 3 entropy-22-01026-t003:** Attack subgraphs generated by the proposed attack graph segmentation algorithm.

Subgraph No.	Number of Vertices	Number of Edges	Complex or Not
1	15	43	N
2	13	37	N
3	13	31	N
4	16	39	N
5	7	25	N
6	9	30	N
7	11	34	N
8	22	57	N
9	6	24	N
10	15	55	N
11	17	58	Y
12	14	40	N
13	10	22	N
14	7	28	N

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
