# Peer review of "Distributed Attack Modeling Approach Based on Process Mining and Graph Segmentation"

_entropy, 2020, doi:10.3390/e22091026_

Round 1

Reviewer 1 Report

This paper presents an algorithm for finding subgraphs in attack graphs discovered from event logs that capture the intrusion behavior. Firstly, a process model is discovered from the event log using heuristic mining algorithm. Then, subgraphs are found and separated from the main process graph. Since each subgraph is smaller in size, it should be easier to humans to understand and analyze it. Heuristic process mining algorithm as well as Alpha miner were implemented in local and distributed environments and tested on real data (security alerts generated by intrusion detection system / intrusion prevention system (IDS/IPS) devices installed in a European Internet service provider between March and August 2016). Also, the segmentation algorithm was tested for different configurations of the distributed environment. The paper also compares performance and memory usage characteristics of discovery and segmentation algorithms. To my opinion, this paper has several serious drawbacks and can be accepted only if significantly improved. See the detailed comments below:

  • The lack of references to other work related to “attack graphs”. For instance, the authors make the following claims without providing any references: “The existing algorithms for attack modeling are generally classified into two categories: alert correlation and alert clustering”; “Process mining can also effectively extract attack patterns from intrusion detection alerts.”  
  • Also, it is not clear why attack graphs should be analyzed by humans while a more robust automatic checking technique can be implemented. The authors say that ”The problem is that such a graph is too complex to be analyzed by humans [3,4]” and it seems references [3] and [4] say nothing about that.
  • One of the main concerns is that this approach is not attack-specific. In fact, it can be applied in different domains.
  • It is not clear why Heuristic mining algorithm was selected for the put forward purposes, because it generates approximate process models that not always precisely present the event log.
  • There are obvious problems with formalization of the approach. The definitions are rather vague. For instance, there is no difference between successor subgraph and independent successor subgraph. This definitions should be given more precisely. Minor comment: Pt = {G,,,} – both parenthesis must be of the same type(“{“,”}”).
  • There is no code/data provided, the results are not reproducible.
  • The meaning of sub-graph extraction is not clear. An independent subgraph can be easily extracted by humans, because it is not connected with other parts of the graph. So, the problem itself is not very clear.

Author Response

Response to Reviewer 1 Comments

Point 1: The lack of references to other work related to “attack graphs”. For instance, the authors make the following claims without providing any references: “The existing algorithms for attack modeling are generally classified into two categories: alert correlation and alert clustering”; “Process mining can also effectively extract attack patterns from intrusion detection alerts.”  

Response 1: We gratefully appreciate for your comment. We have added relevant literature, which are reference [2] and reference [7]  respectively.

Point 2: Also, it is not clear why attack graphs should be analyzed by humans while a more robust automatic checking technique can be implemented. The authors say that ”The problem is that such a graph is too complex to be analyzed by humans [3,4]” and it seems references [3] and [4] say nothing about that.

Response 2: Current models tend to build a large attack graph with numerous vertices and edges to describe all the cyber-attack behaviors. Such a graph is too complex to be analyzed by humans. Although a robust automatic checking technique can be implemented to analyze the attack graph. Manual analysis by the network security administrator is still an important means to ensure network system security in practice. Because an experienced network security administrator can effectively discover the potential danger latent in the attack graph. Furthermore, the network security administrator can carry out a directed and specific action to prevent attack from intruders. References [3] and [4] become [8] and [9]. In these works, researchers tend to build a large attack graph with numerous vertices and edges to describe all the cyber-attack behaviors. It is also mentioned in the literatures that this is the defect of the proposed method. Therefore, to reduce the attack graph complexity, we propose a segmentation algorithm for complex attack graphs.

Point 3: One of the main concerns is that this approach is not attack-specific. In fact, it can be applied in different domains.

Response 3: We appreciate for your valuable comment. Yes, we probably wish our method can be applicable and enables network security administrators to obtain more detailed attack information to make appropriate decisions. It's not a method for a specific attack.

Point 4: It is not clear why Heuristic mining algorithm was selected for the put forward purposes, because it generates approximate process models that not always precisely present the event log.

Response 4: We appreciate for your valuable comment. Alvarenga [27] first applied the heuristic process mining algorithm to attack graph generation and proved the feasibility of the process mining algorithm by experiments. Weijters [39] further proposed the improved heuristic process mining algorithm, which used dependency/frequency table (D/F table) and dependency score to solve the problem of noise and short loops. By using the D/F table, the heuristic process mining algorithm can properly mine the XOR/AND relationships from the intrusion alerts. We follow the work of Alvarenga to generate the initial attack graph. The major work of this paper is to study how to segment the attack graph generated by the heuristic process mining algorithm to make the attack graph more comprehensible to network security administrators. Besides, to make heuristic attack graph mining algorithms adapt to large-scale data, we introduce the distributive algorithm.

Point 5: There are obvious problems with formalization of the approach. The definitions are rather vague. For instance, there is no difference between successor subgraph and independent successor subgraph. This definitions should be given more precisely.

Response 5: Thank you for this valuable feedback. We revised the definition of independent subgraph. First we changed the term of definition 4 to ‘independent successor subgraph’. We also changed the content of definition 4 to clearly describe the characteristics of the independent successor subgraph. For your convenience, the revised definition 4 is as follows:

Definition 4. Independent subgraph. Given a successor subgraph of vertex v, for each vertex in this success subgraph, all the directed edges that end with this vertex should belong to the successor subgraph. Then this subgraph is an independent subgraph with v as the starting vertex.

Point 6: The meaning of sub-graph extraction is not clear. An independent subgraph can be easily extracted by humans, because it is not connected with other parts of the graph. So, the problem itself is not very clear.

Response 6: Thank you for this valuable feedback. To improve the initial attack graphs’ readability, we propose the graph segmentation algorithm to split a complex attack graph into multiple subgraphs while preserving the original structure. The proposed algorithm begins with a search for the branch points from which the subgraphs are split, followed by completion of the subgraphs according to their structure. An incremental update method for the subgraphs is also proposed to adapt to the dynamic change of the attack graph. The proposed algorithm reduces the complexity of the attack graph without ruining its structure facilitating manual analysis by the network security administrator.

Reviewer 2 Report

BRIEF SUMMARY

The authors propose the application of process mining to build attack graphs, i.e. structures representing network intrusion strategies. They present an example based on heuristic miner, as well as an analysis framework for the proposed solution. The obtained results prove that the use of process mining can be a considerable improvement compared to the existing methods.

BROAD COMMENTS.

The paper is well-structured, clear to the reader and provides a good theoretical background of the applied technique, including the necessary literature review and illustrative examples. 

I would suggest providing slightly more literature background about the process mining and reviewing the bibliography - e.g. position 35 - the author's name is actually van der Aalst. Likewise, position 36 seems to be described incorrectly.

Author Response

Response to Reviewer 2 Comments

Point 1: I would suggest providing slightly more literature background about the process mining and reviewing the bibliography - e.g. position 35 - the author's name is actually van der Aalst. Likewise, position 36 seems to be described incorrectly.

Response 1: Thank you for your valuable suggestion. We have supplemented the relevant literature, which are reference [4-6]. We have also revised the bibliography on the original 35 and 36 positions (now 40 and 41).